# Impact of the Aryl Hydrocarbon Receptor on Aurora A Kinase and the G2/M Phase Pathway in Hematopoietic Stem and Progenitor Cells

**Anthony M. Franchini** [1,*], **Keegan L. Vaughan** [1], **Soumyaroop Bhattacharya** [2], **Kameshwar P. Singh** [1], **Thomas A. Gasiewicz** [1] and **B. Paige Lawrence** [1]

1   Department of Environmental Medicine, University of Rochester School of Medicine and Dentistry, Rochester, NY 14642, USA
2   Department of Pediatrics, University of Rochester School of Medicine and Dentistry, Rochester, NY 14642, USA
*   Correspondence: anthony_franchini@urmc.rochester.edu

**Abstract:** Recent evidence suggests that the environment-sensing transcription factor aryl hydrocarbon receptor (AHR) is an important regulator of hematopoiesis. Yet, the mechanisms and extent of AHR-mediated regulation within the most primitive hematopoietic cells, hematopoietic stem and progenitor cells (HSPCs), are poorly understood. Through a combination of transcriptomic and flow cytometric approaches, this study provides new insight into how the AHR influences hematopoietic stem and progenitor cells. Comparative analysis of intraphenotypic transcriptomes of hematopoietic stem cells (HSCs) and multipotent progenitor (MPP) cells from AHR knockout (AHR KO) and wild type mice revealed significant differences in gene expression patterns. Notable among these were differences in expression of cell cycle regulators, specifically an enrichment of G2/M checkpoint genes when *Ahr* was absent. This included the regulator Aurora A kinase (*Aurka*, AurA). Analysis of AurA protein levels in HSPC subsets using flow cytometry, in combination with inducible AHR KO or in vivo AHR antagonism, showed that attenuation of AHR increased levels of AurA in HSCs and lineage-biased MPP cells. Overall, these data highlight a potential novel mechanism by which AHR controls HSC homeostasis and HSPC differentiation. These findings advance the understanding of how AHR influences and regulates primitive hematopoiesis.

**Keywords:** Aurora A kinase; hematopoietic stem cells



## 1. Introduction

Hematopoiesis is the essential process that gives rise to all lineages of blood and immune cells [1]. Self-renewing hematopoietic stem cells (HSCs) are at the apex of this process. HSCs remain predominantly quiescent during an organism's lifespan, but proliferate to replenish lineage committed progenitors [2]. Proximal to HSCs are multipotent progenitor cells (MPPs), which maintain downstream differentiated progenitors of all blood and immune cell populations [3,4]. Collectively, HSCs and MPPs are referred to as hematopoietic stem and progenitor cells (HSPCs). Hematopoietic stem and progenitor cells are dependent on external signals that maintain self-renewal, prompt regulated proliferation, and influence cell programming in order to restock progenitors and downstream cell lineages at steady state and in response to stressors, such as infection or malignancy [5]. This response involves numerous cytokines, growth factors, receptors, and transcription factors [6]. Disrupted regulation of the balance between hematopoietic stem and progenitor cell dormancy and proliferation can have serious long-term consequences [7,8]. Additionally, maintaining stem and progenitor cell quiescence prevents pre-mature stem cell exhaustion [9]. Yet, a critical function of hematopoietic stem and progenitor cells is to rapidly respond to environmental cues. This response involves sensing external signals

that modulate HSCs dormancy versus HSC proliferation and self-renewal, and cue the transition of HSCs into MPPs.

AHR is a member of the Per-ARNT-Sim family, which encompasses a range of environment-sensing transcriptional regulatory factors [10]. Experimental evidence supports a role of AHR in hematopoiesis [11–13]. Recent studies showed changes in functional status and frequency of long-term HSCs and MPPs following loss or antagonism of AHR [14–16]. Other studies have shown that absence of AHR affects downstream lineage committed hematopoietic cells [11,13,16–18]. This includes the observation that the loss of AHR impacts lineage potential of hematopoietic stem and progenitor cells, including biasing towards myeloid-biased lineage precursors [16]. In other studies, antagonism of the AHR in human HSCs supports a role in regulating HSC proliferation and differentiation [19]. However, the genes influenced by AHR in the context of regulating HSCs and MPP cells (i.e., hematopoietic stem and progenitor cells) have yet to be fully elucidated.

The study presented here was undertaken to better define the role of AHR in regulating hematopoietic stem and progenitor cells, and to identify genes and pathways influenced by AHR during the earliest stages of hematopoiesis. Using a combination of transcriptional and flow cytometric approaches, we present evidence that AHR shapes the HSPC transcriptome and differentiation program, influencing processes and pathways involved in critical checkpoints of the cell cycle, including Aurora A kinase. These observations expand the role of AHR as a central regulator of primitive stages of hematopoiesis.

## 2. Materials and Methods

### 2.1. Mice and In Vivo Treatment

C57BL/6J wild type mice were purchased from the Jackson Laboratory (Bar Harbor, ME, USA). Initial breeding stocks of B6.129-Ahr$^{tm1Bra}$/J (AHR KO) and Ahr$^{tm3.1Bra}$/J (AHR$^{Fx/Fx}$) mice were provided by Christopher Bradfield (University of Wisconsin, Madison, WI, USA) [20]. B6.129-Gt(ROSA)26Sortm1(Cre/ERT2)Tyj/J (CRE$^{ERT2}$) mice were purchased from the Jackson Laboratory [21] and crossed with AHR$^{Fx/Fx}$ mice to generate AHR$^{Fx/Fx}$CRE$^{ERT2}$ mice [16]. All data presented are from female mice that were 6–10 weeks of age at the time of experiments. Excision of *Ahr* from tamoxifen-treated AHR$^{Fx/Fx}$ CRE$^{ERT2}$ (AHR iKO) mice was confirmed by PCR [16,20]. All primers used for genotyping PCR can be found in Supplemental Table S1.

To assess in vivo proliferation, mice were administered 120 mg bromodeoxyuridine (BrdU, Sigma-Aldrich, St. Louis, MO, USA) per kg body weight by intraperitoneal (i.p.) injection 2 h prior to the termination of the experiment. For some studies, AHR$^{Fx/Fx}$ and AHR$^{Fx/Fx}$CRE$^{ERT2}$ mice were administered 25 mg tamoxifen/kg body weight by i.p. injection on 3 consecutive days [16,21]. After the third dose of tamoxifen, mice were allowed to rest for 14 days prior to assessment. Tamoxifen was purchased from Sigma-Aldrich (St. Louis, MO, USA). For AHR antagonism, CH-223191 (Sigma-Aldrich) was dissolved in corn oil at a concentration of 0.5 mg/mL, and 100 μg per mouse was delivered by i.p. injection [13,22,23]. All animal treatments had prior approval of the Institutional Animal Care and Use Committee of the University of Rochester. The University is accredited by the Association for Assessment and Accreditation of Laboratory Animal Care (AAALAC). Animals were treated humanely and with due consideration to alleviation of distress and discomfort, following U.S. Public Health Service Policy on Human Care and Use of Laboratory Animals guidelines for the handling of vertebrate animals.

### 2.2. Collection of Bone Marrow Cells

Both femurs and tibias were excised from each mouse, cleared of adherent tissue, and crushed in mortar and pestle to release the bone marrow cells [11,15,18,24]. Bone marrow cells were resuspended in Iscove's Modified Dulbecco's Medium (IMDM, Gibco, Gaithersburg, MD, USA) supplemented with 2.5% fetal bovine serum (HyClone, Logan, UT, USA). Erythrocytes were lysed using an ammonium chloride solution (0.15 M NH$_4$Cl, 10 mM NaHCl$_2$, 1 mM Na$_2$EDTA) for five minutes at room temperature, and bone marrow

cells were passed through 40 μm filters twice to remove debris. Cells were immediately used for analysis by flow cytometry, or subjected to further purification and extraction of cellular materials.

### 2.3. HSPC Isolation and Flow Cytometry

For cell sorting prior to RNA isolation, lineage positive (CD5, CD45R, CD11b, Gr-1, 7-4, Ter-119) bone marrow cells were removed by positive selection using immuno-magnetic beads (Miltenyi, Waltham, MA, USA) prior to labeling the remaining cells for fluorescence activated cell sorting (FACS). Lineage negative cells were incubated with anti-CD16/32 to block non-specific binding, and labeled with antibodies against Sca1, cKit, CD135, CD48, CD150, as well as a cocktail of lineage-specific antibodies to identify MPPs (Lin$^-$CD135$^-$ckit$^+$Sca1$^+$CD48$^+$CD150$^{+/-}$) and HSCs (Lin$^-$CD135$^-$ckit$^+$Sca1$^+$CD48$^-$CD150$^+$). Sorting of HSC and MPP populations was performed using a BD FACSAriaII flow cytometer (BD Biosciences, San Jose, CA, USA) in the University of Rochester Flow Cytometry Core. The gating strategy used for sorting HSCs and MPPs is presented in Supplemental Figure S1. Details regarding the antibodies used are provided in Supplemental Table S2.

To identify distinct HSPC populations using analytical flow cytometry, bone marrow cells were incubated with fluorescently tagged antibodies that recognize the following cell surface markers: CD117, CD48, CD34, CD135, CD48, CD150 and lineage markers (CD3, CD45R, CD11b, Ly6G, and TER119) [25]. Non-specific binding was blocked by pre-incubation of cells with anti-CD16/32. After cell surface labeling, cells were fixed using 2% paraformaldehyde and analyzed directly by flow cytometry, or permeabilized for assessment of intracellular molecules. To examine Aurora A kinase, cells were permeabilized using 1% saponin (Sigma-Aldrich) prior to incubation with a polyclonal anti-Aurora A antibody (ST46-07, Novus Biologicals, Littleton, CO, USA), in combination with allophycocyanin conjugated-donkey anti-rabbit IgG antibody (Biolegend). To detect BrdU incorporation, cells were permeabilized with Triton X-100 (0.5%, Sigma-Aldrich, St. Louis, MO, USA) and incubated with a directly conjugated anti-BrdU monoclonal antibody. A list of all antibodies, including fluorochrome, vendor, and amount used is in Supplemental Table S2. Fluorescence minus one (FMO) controls were used to determine non-specific fluorescence and define all gating parameters [26]. Two to three million bone marrow cells from individual mice were stained, and 1 million events were collected using a LSRII flow cytometer (BD Biosciences, San Jose, CA, USA). Data were analyzed using the FlowJo software program (Version 10, TreeStar, Ashland, OR, USA). The specific combinations of molecular markers used to identify cell populations are denoted in the figure legends. The gating strategy used to identify HSPC subsets (i.e., HSCs, MPP1, MPP2, MPP3 and MPP4 cells) is presented in Supplemental Figure S2.

### 2.4. RNA Sequencing Library Construction and Transcriptomic Analysis

RNA was isolated from sorted HSCs and MPPs using RNeasy Mini kits (Qiagen, Valencia, CA, USA). The concentration of RNA was determined using an Agilent 2100 bioanalyzer (Agilent Technologies, Santa Clara, CA, USA). RNA (1 ng) was amplified using SMARTer library Ultra Low cDNA v4 kit (Takara Bio, Mountainview, CA, USA). Sequencing cDNA libraries were preparing using a NexteraXT DNA library prep kit (Illumina, San Diego, CA, USA). The cDNA library (150 pg/sample) was sequenced on an Illumina HiSeq2500 system to generate ~20 million 100-bp single end reads per sample. Sequences were aligned against the mouse genome mm10 using the Splice Transcript Alignment to a Reference (STAR) algorithm [27], counted with HTSeq [28], and normalized for total counts (counts per million, CPM). A non-specific filtering strategy was used to remove genes with low expression values. One of the samples of MPP data from wildtype mice was removed from differential expression and other downstream analysis because it had much lower mapping quality control metrics. This was further verified by the single dataset being significantly different from all other samples in principal component analysis (PCA) and unsupervised hierarchical clustering using all genes. Differential

gene expression was assessed by paired DE-Seq2 [29] to identify genes with significant differences in mean expression (false discovery rate, FDR, <0.05). Differentially expressed genes (DEGs) identified by direct comparison of datasets from HSCs (across genotype) and MPPs can be found in Supplemental Tables S3 and S4, respectively. DEGs identified by intraphenotypic analysis (i.e., HSC vs. MPP within genotype) are shown in Supplemental Tables S5 and S6, respectively. Genes identified as differentially expressed in comparisons were submitted to upstream regulator and canonical pathways analysis using Ingenuity Pathway Analysis (IPA, Qiagen, Redwood City, CA, USA) [30]. Canonical pathways identified by intraphenotypic analysis (i.e., HSC vs. MPP within genotype) are shown in Supplemental Tables S7 and S8, respectively.

### 2.5. Statistical Analyses

With the exception of sequencing data, statistical analyses were performed using JMP Pro 14.0.0 (SAS Institute, Cary, NC, USA). Differences between means of multiple independent variables were compared using one-way ANOVA followed by post hoc tests (Tukey honestly significant difference or Dunnett's test). Differences between two groups at a single point in time were analyzed using Student's *t* test. The slope of the line was calculated using goodness of fit modeling and derivation from non-zero slope determination was performed. Differences in mean values were considered statistically significant when *p*-values were < 0.05. Error bars on all graphs represent the SEM. Linear regression was performed using Prism (Version 8, GraphPad, San Diego, CA, USA). $R^2$ and *p*-value of the correlation were calculated using a goodness of fit model and compared to a non-zero slope are shown for each regression analysis.

### 2.6. Data Availability

RNA-sequencing data have been deposited to the NIH Gene Expression Omnibus (GEO) under accession number (GSE163284).

## 3. Results

### 3.1. Absence of AHR Alters HSPC Transcriptome

Isolated bone marrow cells from AHR KO mice have a greater frequency of HSCs compared to bone marrow cells from wild type mice (Figure 1A). Moreover, a higher proportion of HSCs were proliferating. Specifically, there was a 28% increase in BrdU+ HSCs in AHR KO mice compared to HSCs from wild type mice (Figure 1B). The proliferative capacity of HSCs is integrally linked with their transcriptional program, and with cues that drive their differentiation to multipotent progenitors (MPP) [6,31,32]. Compared to wild type mice, AHR KO mice have about two-fold higher percentage of MPP in the bone marrow (Figure 1C). However, unlike HSCs, the proportion of MPP that were BrdU+ was not statistically significantly different between wild type and KO mice (Figure 1D). However, when the proportion of MPP sub-populations that were BrdU+ was examined, there were significantly more BrdU+ MPP1 cells in AHR KO mice, comparted to MPP1 cells from wild type mice (Supplemental Figure S3).

Given that AHR is a transcription factor, it is logical to predict that absence of AHR is influencing HSC frequency, proliferation, and transition to MPP via alterations in their transcriptional program. However, the full extent by which AHR influences the transcriptome of HSCs and MPPs is incomplete, as prior studies did not separate HSCs from MPPs (i.e., prior studies used bulk hematopoietic stem and progenitor cells) or used a limited microarray to assess gene expression differences [11,17]. To directly address this, we evaluated the transcriptome of sorted HSCs and MPPs. Specifically, HSCs and MPPs from wild type and AHR KO mice were isolated using FACS followed by high throughput RNA sequencing (RNA-Seq; Figure 1C). Inter-sample Euclidean distances confirmed delineation of HSCs and MPPs into separate clusters (Figure 1E). HSCs and MPPs clustered more closely within the same genotype (Figure 1E). Comparison of gene expression profiles in HSCs from wild type and AHR KO mice revealed 103 differentially expressed genes (DEGs; Figure 1F, and

Supplemental Table S3). In contrast, comparison of expressed genes in MPPs from wild type and AHR KO mice revealed only six DEGs. Notably, *Akr1c13* was identified in direct comparisons of AHR KO versus wild type HSCs and MPPs (Supplemental Tables S3 and S4). Among these, only *Garnl3* was shared with DEGs from HSCs, and the other five were distinct from the DEGs in HSCs (Figure 1G, and Supplemental Table S3). Among these six genes in MPPs, two encode receptors (*Ffar2* and *Slamf6*), two encode proteins involved cell metabolism (*Akr1c13*, *Tmem18*; Supplemental Table S4), and two are for factors involved in cell signaling (*Garnl3*, *Mrvi1*; Supplemental Table S4).

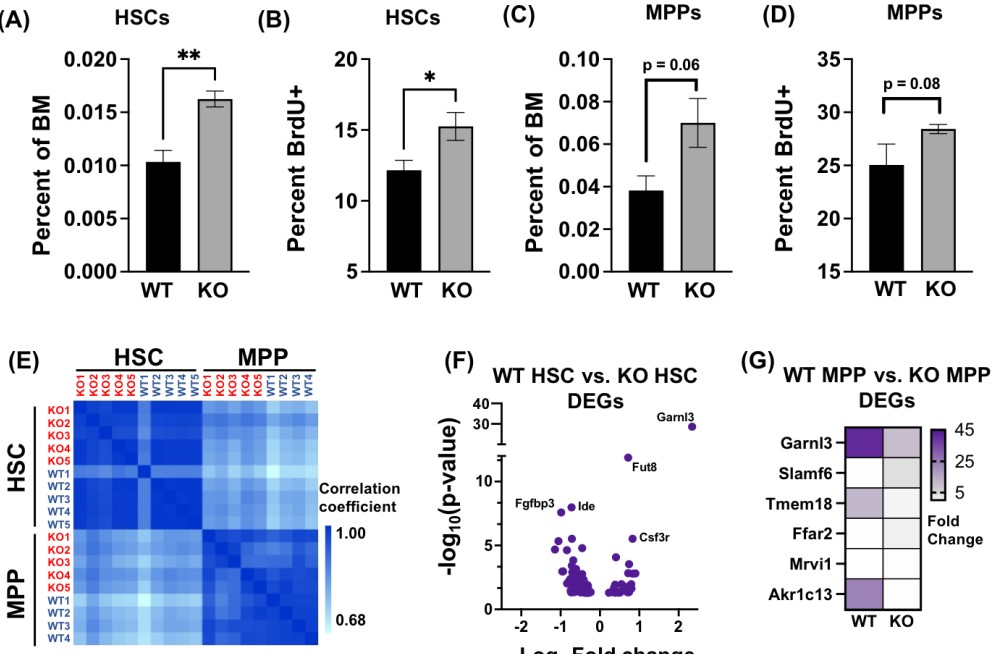

**Figure 1.** Proliferation Gene expression patterns in HSC and MPP from wild type and AHR KO mice. Bone marrow (BM) cells were isolated from wild type C57Bl/6 (WT) and global constitutive AHR knockout mice (KO). The percentage of (**A**) HSCs and (**B**) BrdU+ HSCs in WT and AHR KO mice was determined by flow cytometry (n = 5 mice per genotype). Percentage of (**C**) MPPs in the bone marrow and (**D**) percentage of BrdU+ MPPs were determined by flow cytometry. HSC were defined as Lin$^-$CD135$^-$CD117$^+$Sca1$^+$CD48$^-$CD150$^+$ cells and MPPs were defined as Lin$^-$CD135$^-$CD117$^+$Sca1$^+$CD48$^+$CD150$^{+/-}$ cells. BrdU was delivered (i.p.) 2 h prior to bone marrow cell isolation. Asterisks denote *p* < 0.05 between genotype; Student's *t*-test. (**E–G**) HSCs and MPPs from individual C57BL/6 (WT) mice and AHR KO mice were isolated by cell sorting (FACS) followed by RNA-Seq. HSCs were defined as Lin$^-$CD135$^-$CD117$^+$Sca1$^+$CD48$^-$CD150$^+$ cells, and MPPs were defined as Lin$^-$CD135$^-$CD117$^+$Sca1$^+$CD48$^+$CD150$^{+/-}$ cells. The gating strategy for sorting is in Supplemental Figure S1. (**E**) Heatmap of the hierarchically clustered Euclidean distances between samples from the regularized log transformation of the normalized count data shows sample grouping by cell types. Darker shades of blue indicate sample distance and higher relatedness between libraries. (**F**) Volcano plot depicts the 103 differentially expressed genes (DEGs) identified in HSCs from WT and AHR KO mice. The complete list of HSC DEGs, with fold-change and *p*-value, can be found in Supplemental Table S3. (**G**) Heatmap of average normalized counts of the six genes that were differentially expressed in MPPs from WT and AHR KO mice. Gene expression by fold-change is shown. The complete list of MPP DEGs, with fold-change and *p*-value, can be found in Supplemental Table S4. The error bars indicate the SEM. Asterisks (*) denote *p* < 0.05 by Student's *t*-test. Double asterisks (**) denote *p* < 0.01 by Student's *t*-test.

To further understand how AHR influences the transcriptional profile of hematopoietic stem and progenitor cells, we compared gene expression in HSCs vs. MPPs within the same genotype, and compared the transcriptional landscape in the presence and absence of

AHR. Between the wild type HSC and MPP transcriptomes, 1586 differentially expresses genes were identified (Figure 2A, Supplemental Table S5). Between the AHR KO HSC and MPP transcriptomes, 2474 DEGs were identified (Figure 2A, Supplemental Table S6). Thus, when AHR was absent, there were 55% more DEGs identified between these two populations of cells compared to when AHR was present. Comparative analysis of these intraphenotypic datasets (i.e., comparing the DEGs between HSCs and MPPs from wild type mice to DEGs among HSCs and MPPs from AHR KO mice) revealed that that they had 1124 genes in common. In contrast, 1315 DEGs were unique to the AHR KO HSC to MPP intraphenotypic comparison, while 460 genes were uniquely different compared to the wild type HSC to MPP dataset (Figure 2A). This two-way comparative approach indicates that absence of AHR has a pronounced impact on the activation and differentiation of hematopoietic stem and progenitor cells.

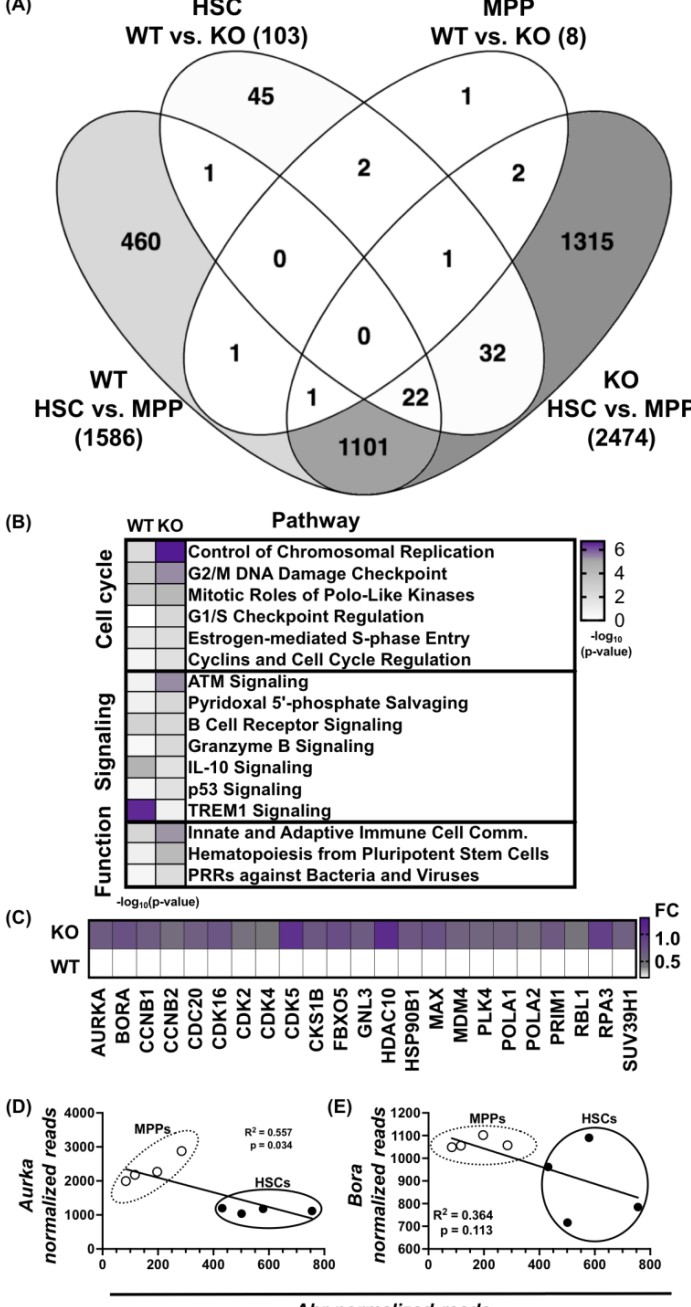

**Figure 2.** Absence of AHR alters cellular programming of hematopoietic stem and progenitor cells. The transcriptomes of phenotypically defined HSCs (Lin⁻CD135⁻CD117⁺Sca1⁺CD48⁻CD150⁺) and

MPPs (Lin⁻CD135⁻CD117⁺Sca1⁺CD48⁺CD150⁺) cells were compared directly and across genotypes. (**A**) Four-way Venn diagram depicts the number of differentially expressed genes (DEGs) in the following comparisons: WT HSC vs. WT MPP, KO HSC vs. KO MPP, WT HSC vs. KO HSC, and WT MPP vs KO MPP. The number in parentheses indicates the total number of DEGs in each of these comparisons. The complete lists of DEGs from these comparisons are in Supplemental Tables S5 and S6. (**B**) Pathways identified as significantly different in the WT and KO intraphenotypic datasets (i.e., comparing DEGs in HSCs vs. MPPs across genotype). The heatmap denotes $-\log_{10}(p$ value) in WT and KO intraphenotypic HSPC datasets (numerical data are in Supplemental Tables S7 and S8). (**C**) Heatmap depicts the $\log_2$ fold change of cell cycle related DEGs identified in the six cell cycle pathways identified (i.e., pathways in panel B). The fold-change and *p*-value for each DEG can be found in Supplemental Table S4. (**D**,**E**) Linear regression analysis was performed on the normalized reads of *Aurka* and *Bora* against *Ahr*. $R^2$ and *p*-values are shown on each graph. Solid (HSC) and open (MPP) circles denote data from individual samples. The numerical information represented in each plot is Supplemental Tables S3 and S4.

Pathways analysis was utilized to compare and relate changes to the transcriptional landscape in order to identify functional clusters and cellular pathways affected by the absence of AHR. Comparison of the DEGs in the wild type and KO intraphenotypic datasets identified 37 unique pathways attributed to the lack of AHR. The most affected pathways, ranked by *p*-value, were related to cell cycle, signaling, and stem cell function (Figure 2B, Supplemental Tables S7 and S8). Within the six cell cycle pathways that were significantly different, 23 DEGs were unique to the AHR KO intraphenotypic dataset (Figure 2C). That is, these genes were not differentially expressed in the HSC-to-MPP comparison from wild type mice, but were differentially expressed between HSCs and MPPs from AHR KO mice. Notable within the DEGs was polo-like kinase and G2/M phase regulator Aurora A kinase (*Aurka*, AurA), which exhibited a 1.6-fold increase in gene expression (0.67 $\log_2$ fold change) in the KO dataset (Figure 2C). In addition to increased *Aurka*, absence of AHR correlated with greater expression of the gene for its dimerization partner, *Bora*, which was 1.7-fold-higher in the AHR KO data set, compared to wild type dataset (Figure 2C, Supplemental Tables S3 and S4). Furthermore, there was a statistically significant inverse correlation between *Aurka* and *Ahr* levels within the HSC-to-MPP comparison from wild type mice (Figure 2D). That is, compared to HSCs, bulk MPPs expressed higher levels of *Aurka* and lower *Ahr* levels. *Bora* expression levels showed a similar pattern, although in HSCs, *Bora* expression was more variable than *Aurka* (Figure 2E). These findings suggest that AHR influences expression of key G2/M phase checkpoint regulators in hematopoietic stem and progenitor cells.

### 3.2. Acute Loss or Antagonism of AHR Results in Increased AurA in Hematopoietic Stem and Progenitor Cells

To further assess the connection between AHR and Aurora A kinase, the relative levels of AurA protein in hematopoietic stem and progenitor cells were examined using flow cytometry. Given that stem and progenitor cells are relatively infrequent cells, this approach has a distinct advantage over immunoblotting in that AurA protein can be examined without pooling cells from large numbers of mice. Consistent with AurA's ubiquitous expression in other cell types [32], AurA was detected in over 99% of all hematopoietic stem and progenitor cells (Figure 3A). HSCs and all MPP subpopulations expressed AurA, with no discernable difference in the percentage of AurA+ cells among HSCs or MPPs (Figure 3B). Furthermore, regardless of their lineage biases and phenotype, the mean fluorescence intensity (MFI) of AurA was not significantly different between HSCs and any of the subpopulations of MPPs (Figure 3C), indicating similar levels of expression amongst hematopoietic stem and progenitor cell subsets.

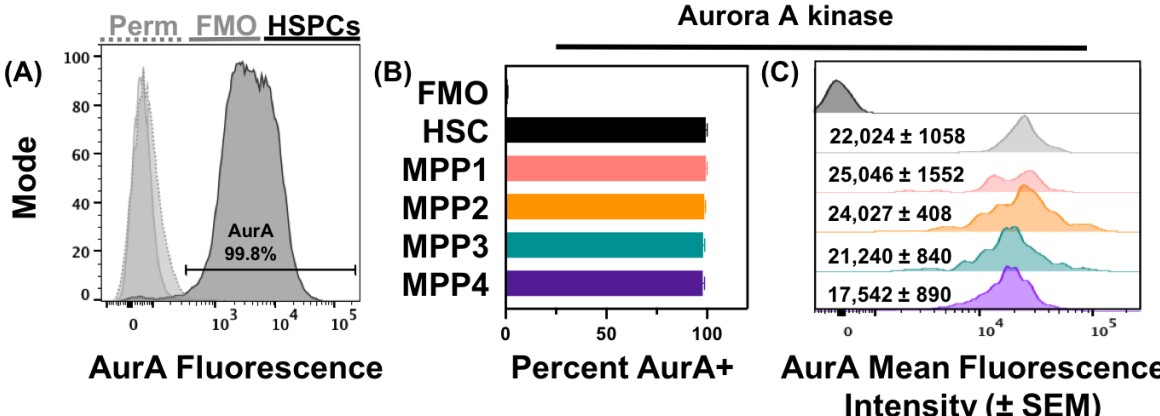

**Figure 3.** Aurora A kinase protein levels are similar in HSPC subsets. Bone marrow cells from C57Bl/6 mice (n = five) were isolated and stained for flow cytometry. (**A**) The dark grey histogram depicts AurA fluorescence in hematopoietic stem and progenitor cells. Fluorescence minus one (FMO) and permeabilization controls are shown in light grey solid and dotted grey lines, respectively. (**B**) Graph depicts the percentage of AurA+ HSPC subsets. Error bars denote SEM. Subsets were defined as follows: HSC (Lin$^-$Sca1$^{high}$CD117$^+$CD34$^-$CD135$^-$CD48$^-$CD150$^+$), MPP1 (Lin$^-$Sca1$^{high}$CD117$^+$CD34$^+$CD135$^-$CD48$^-$CD150$^+$), MPP2 (Lin$^-$Sca1$^{high}$CD117$^+$CD34$^+$CD135$^-$CD48$^+$CD150$^+$), MPP3 (Lin$^-$Sca1$^{high}$CD117$^+$CD34$^+$CD135$^-$CD48$^+$CD150$^-$), and MPP4 cells (Lin$^-$Sca1$^{high}$CD117$^+$CD34$^+$CD135$^+$CD48$^-$CD150$^-$). (**C**) Histograms show the mean fluorescence intensity (MFI) of AurA in each HSPC subset. The numbers on the graph indicate the mean MFI (±SEM) of AurA in each indicated sub-population for all samples. There were five mice used, and bone marrow cells from individual mice were not pooled. A detailed gating strategy is provided in Supplemental Figure S2.

To determine if the absence of AHR alters AurA levels in hematopoietic stem and progenitor cells, AHR$^{Fx/Fx}$ mice were crossed with Cre$^{ERT2}$ mice, to generate AHR$^{Fx/Fx}$Cre$^{ERT2}$ mice [16]. Two weeks after tamoxifen treatment, *Ahr* was confirmed to be excised in bone marrow cells by PCR in these inducible AHR knockout (AHR iKO) mice, whereas *Ahr* was not excised in tamoxifen-treated AHR$^{Fx/Fx}$ mice (Figure 4A). Inducible ablation of AHR resulted in a 20% increase in AurA protein in the hematopoietic stem and progenitor cells of AHR iKO mice, compared to hematopoietic stem and progenitor cells from control AHR$^{Fx/Fx}$ mice (Figure 4B,C). Compared to AHR$^{Fx/Fx}$ controls, AurA levels were generally higher among individual HSPC subsets from AHR iKO mice, but not statistically different in HSCs (Figure 4D). Within the MPP subsets, statistically higher levels of AurA were detected in MPP1 and MPP3 cells (Figure 4D–H) compared to AHR$^{Fx/Fx}$ controls. These findings indicate that acute AHR deletion was sufficient to alter the level of AurA within hematopoietic stem and progenitor cells, with main effects occurring within MPPs.

The AHR antagonist CH-223191 was used to further examine the relationship between AHR attenuation and AurA levels. Administration of CH-223191 did not alter the frequency of hematopoietic stem and progenitor cells in the bone marrow (Figure 5A), nor did it affect the percentage of hematopoietic stem and progenitor cells that were AurA+ (Figure 5B). However, hematopoietic stem and progenitor cells from mice treated with CH-223191 had approximately 30% higher levels of AurA compared to hematopoietic stem and progenitor cells from mice given the vehicle control (Figure 5C). Similarly, AHR antagonism triggered a 10–25% elevation in AurA levels in HSCs and in MPP subsets compared to cells from controls (Figure 5D–H). Thus, similar to inducible deletion of AHR, antagonism of AHR increased AurA levels within HSPC subpopulations.

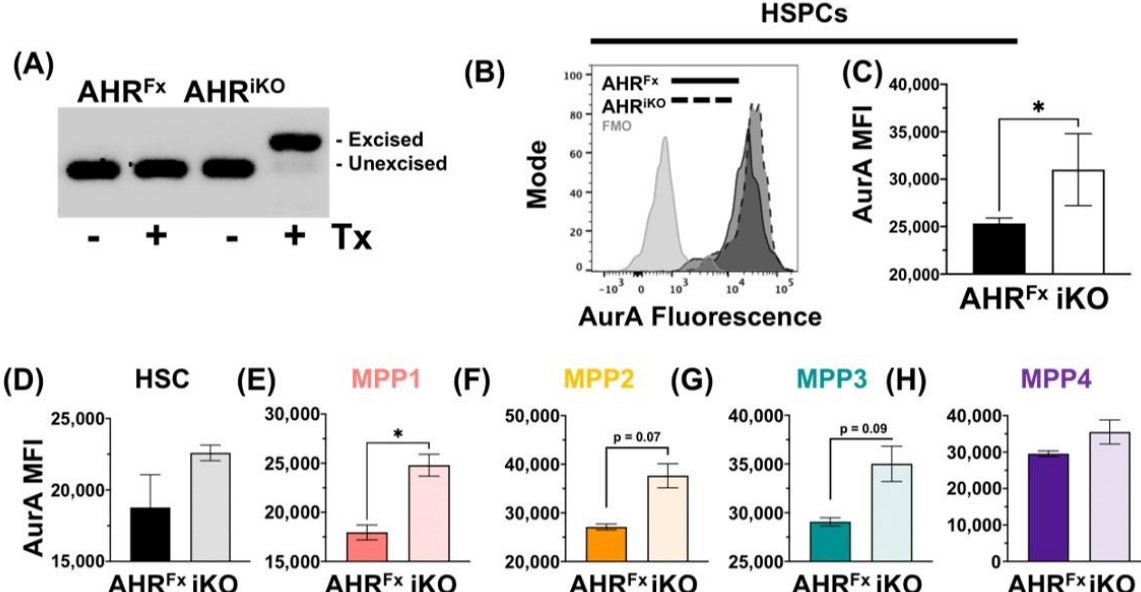

**Figure 4.** Inducible deletion of AHR increases AurA expression in hematopoietic stem and progenitor cells. AHR$^{Fx/Fx}$ and AHR$^{CreERT2}$ mice (five mice per genotype) were administered tamoxifen i.p., 25 mg/Kg body weight (BW) daily for three consecutive days. Two weeks after final tamoxifen (Tx) treatment, bone marrow cells from AHR$^{Fx/Fx}$ and inducible AHR knockout (AHR iKO; iKO) mice were isolated and *Ahr* gene excision assessed using PCR. (**A**) Endpoint PCR run on an agarose gel shows the amplified DNA obtained from AHR$^{Fx/Fx}$ mice with and without Tx treatment, and from AHR$^{Fx/Fx}$Cre$^{ERT2}$ mice with and without Tx treatment (labeled AHR iKO). No pooling of cells from mice was used (i.e., PCR is performed on samples obtained from an individual mouse, and gel shows one mouse from each group). (**B**) Representative histogram depicts AurA fluorescence in hematopoietic stem and progenitor cells (HSPCs, Lin$^-$CD117$^+$Sca-1$^+$ cells) from AHR$^{Fx/Fx}$ (solid line) and AHR iKO (dotted line) mice (all mice received Tx). The fluorescence minus one (FMO) control is shown in light grey. (**C**) The graph shows the AurA mean fluorescence intensity (MFI) in hematopoietic stem and progenitor cells from AHR$^{Fx/Fx}$ (AHR$^{Fx}$) and AHR iKO mice (iKO). (**D–H**) Graphs depict the mean AurA MFI in (**D**) HSCs, (**E**) MPP1, (**F**) MPP2, (**G**) MPP3, and (**H**) MPP4 cells. Cell subsets were defined as follows (gating strategy in Supplemental Figure S2): HSC (Lin$^-$Sca1$^{high}$CD117$^+$CD34$^-$CD135$^-$CD48$^-$CD150$^+$), MPP1 (Lin$^-$Sca1$^{high}$CD117$^+$CD34$^+$CD135$^-$CD48$^-$CD150$^+$), MPP2 (Lin$^-$Sca1$^{high}$CD117$^+$CD34$^+$CD135$^-$CD48$^+$CD150$^+$), MPP3 (Lin$^-$Sca1$^{high}$CD117$^+$CD34$^+$CD135$^-$CD48$^+$ CD150$^-$), and MPP4 cells (Lin$^-$Sca1$^{high}$CD117$^+$CD34$^+$CD135$^+$CD48$^-$CD150$^-$). The error bars indicate the SEM. Asterisks (*) denote $p < 0.05$ by Student's *t*-test.

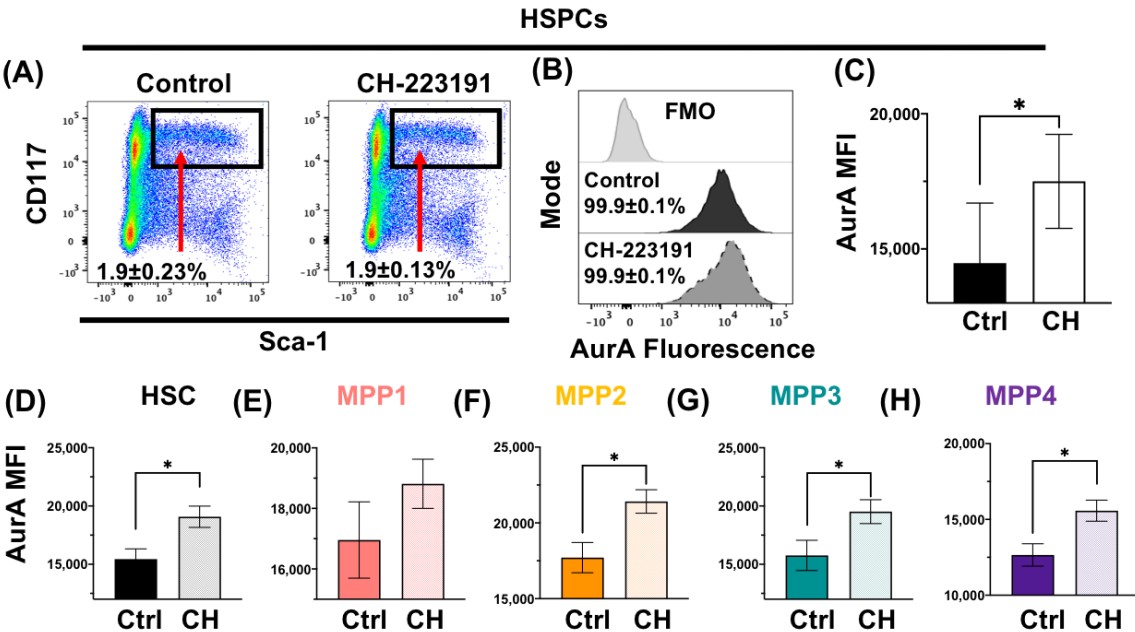

**Figure 5.** In vivo AHR antagonism enhances AurA expression in hematopoietic stem and progenitor cells. C57Bl/6 mice were administered a single dose of CH-223191 (100 µg; CH) or corn oil vehicle control (Ctrl) by i.p. injection (five mice per group). Bone marrow cells were harvested 2 days later for analysis using flow cytometry. (**A**) FACS plots depict hematopoietic stem and progenitor cells (Lin$^-$CD117$^+$Sca$^-$1$^+$) from control or CH treated mice. The number on each plot denotes the mean percentage of hematopoietic stem and progenitor cells ($\pm$SEM) in bone marrow. (**B**) Representative histograms depict AurA staining of hematopoietic stem and progenitor cells from Ctrl (black fill) and CH-treated mice (dark grey fill). The fluorescence minus one (FMO) control is shown in light grey. The numbers indicate the mean percentage ($\pm$SEM) of AurA-positive cells in each group. (**C**) Graph shows the MFI of AurA in hematopoietic stem and progenitor cells from Ctrl and CH treated mice. (**D–H**) Graphs depict the mean AurA MFI in (**D**) HSC, (**E**) MPP1, (**F**) MPP2, (**G**) MPP3, and (**H**) MPP4 cells. Cell subsets were defined as follows: HSC (Lin$^-$Sca1$^{high}$CD117$^+$CD34$^-$CD135$^-$CD48$^-$CD150$^+$), MPP1 (Lin$^-$Sca1$^{high}$CD117$^+$CD34$^+$CD135$^-$CD48$^-$CD150$^+$), MPP2 (Lin$^-$Sca1$^{high}$CD117$^+$CD34$^+$CD135$^-$CD48$^+$CD150$^+$), MPP3 (Lin$^-$Sca1$^{high}$CD117$^+$CD34$^+$CD135$^-$CD48$^+$CD150$^-$), and MPP4 cells (Lin$^-$Sca1$^{high}$CD117$^+$CD34$^+$CD135$^+$CD48$^-$CD150$^-$). The error bars indicate the SEM. Asterisks (*) denote $p < 0.05$ by Student's *t*-test.

## 4. Discussion

Hematopoietic stem and progenitor cells are responsible for providing all the cells of the blood and immune system across the entire lifetime. A key aspect of regulating hematopoiesis and hematopoietic stem and progenitor cell function is balancing stem cell dormancy with the need to proliferate and differentiate into progenitor cells in response to external signals. Understanding the changes that occur within the HSC and MPP populations that are governed by AHR is critical to fully understanding the foundational biology of this critically important pool of hematopoietic progenitor cells. The current study supports the idea that AHR regulation of hematopoietic stem and progenitor cells is very complex, context specific, and that AHR influences their proliferation and differentiation via distinct mechanisms.

With regard to proliferation, transcriptional analyses indicate that AHR interacts with multiple cell cycle pathways, including the G1/S and G2/M phase checkpoint pathways, which reinforces the idea that AHR is a regulator of the genes that control stem and progenitor cell proliferation [31,33–38]. This idea aligns well with prior reports in embryonic stem cells, which showed that AHR must be repressed during the outset of stem cell mitotic

differentiation, and that AHR expression disrupts stem cell cycle progression [39,40]. The current study expands the role of AHR as a regulator of cycle progression, particularly factors associated with the G2/M phase transition in hematopoietic stem and progenitor cells. For example, the absence of AHR correlated with altered expression of *Ccnb1*, *Ccnb2*, *Cdkn1a,* and *Mdm4* (Supplemental Tables S3 and S4) non-hematopoietic cells [38,41–43]. In addition, the current study reveals a novel association between AHR and AurA, which is particularly interesting because AurA and other Polo-like kinases play critical roles in mitosis [44–46]. Mechanistically, it is plausible that AHR regulates expression of AurA by directly binding DNA at AHR responsive elements (AHRE) upstream of the transcriptional start site, acting as a transcriptional regulator or enhancer [47,48]. Direct AHR binding to the upstream regulatory region of the AurA gene could also regulate gene expression by chromatin loop formation, such as has been shown for the gene *Nanog* [49]. Furthermore, given that the association between AHR and *Aurka* was observed in the context of AHR knockout or antagonism, it is possible that AHR acts as an insulator, or that AHR is controlling expression of other factors that directly control AurA expression. Overall, loss and antagonism of AHR increased intracellular levels of AurA, which is novel and consequential because it suggests a potential AHR-mediated mechanism for regulating the earliest stage of hematopoiesis.

In addition to loss of AHR, AHR antagonism affected AurA levels in hematopoietic stem and progenitor cells, and this is consistent with prior reports that AHR antagonism affects the frequency of MPP cells and HSC proliferation [16,19]. While several prior studies indicate AHR influences hematopoiesis [12,16,17,50], there has been less focus on the role of AHR in the regulation of the balance between HSC and MPPs. This study suggests that AHR mediates aspects of this balance. For instance, it is plausible that the elevated AurA observed when AHR is attenuated creates a more permissive cellular state leading to elevated proliferation. Dysregulation in AurA, as well as other cell cycle machinery, can influence the proliferative capacity of hematopoietic stem and progenitor cells, accelerate cellular senescence, and contribute to development of hematological diseases, such as myelodysplastic syndromes and leukemia [51–54]. For example, increased levels of AurA align with elevated proliferation in murine fibroblasts, mammary epithelia, and squamous carcinoma cells [55–57]. Moreover, chronic myeloid leukemia cells express significantly lower amounts of *Ahr* transcript compared to healthy controls [58]. By influencing HSC and MPPs, AHR may also affect early signals that impact the lineage that hematopoietic stem and progenitor cells are more likely to become (i.e., lineage bias). Recent work by Smith et al. [12] and Vaughan et al. [16] showed that AHR influences hematopoietic stem and progenitor cell blood cell lineage bias. Given that proliferation and differentiation are tightly linked in hematopoietic stem and progenitor cells [3,6,8,24,25], AHR mediated regulation of AurA and other cell cycle factors could be one mechanism by which AHR contributes to this lineage bias. Further support for a connection between AHR and cell cycle regulation with hematopoietic stem and progenitor cells comes from observations that absence of AHR correlates with a higher percentage of proliferating HSCs in mice [16], and that inhibition of AHR promotes HSC proliferation in vitro [19,59]. Beyond the hematopoietic niche, AHR has been observed to regulate cell cycle entry and progression in cardiac, embryonic, epidermal, liver, and thymus stem cells [60–65]. Thus, modulation of AHR may influence signals that regulate proliferation and differentiation of hematopoietic stem and progenitor cells early-stage hematopoiesis.

Further support for the idea that AHR influences early stages of hematopoiesis is that the majority of transcriptomic differences related to the absence or presence of AHR were uncovered when comparing the transcriptional landscape of hematopoietic stem and progenitor cells, rather than directly comparing gene expression profiles within the same cell population. Herein, these data suggest that AHR may regulate factors important for the transition of HSC to MPPs, rather than controlling a small subset of factors. The limited number of DEGs detected when comparing HSCs in wild type and AHR KO mice may also reflect the fact that a large portion of the HSC pool is transcriptionally dormant [66]. The

limited number of DEGs in MPPs from wild type versus AHR KO mice could simply reflect the heterogenous nature of MPPs [24]. Further separation of distinct MPP subsets will be needed to examine the extent and specificity of AHR on their individually specialized transcriptomes. Regardless, when considered in the context of the rich set of DEGs in the intraphenotypic datasets (i.e., comparing DEGs in HSCs vs. MPPs within and across genotype), transcriptome profiling indicates that AHR plays important role in regulation of the transition of HSCs into MPPs. The distinct transcriptional profiles across HSCs and MPPs also suggest that they may be poised to respond differently to external cues. This may help to explain associations of attenuation of AHR and lineage biasing of hematopoietic cells that have been reported [12,16,17].

The ability to modulate hematopoietic stem and progenitor cells via AHR has the potential to alter hematopoiesis in ways that can be beneficial or deleterious. Hematopoiesis is a complex process wherein small alterations have significant downstream consequences on host defenses against infection, repair following injury, hematologic diseases, as well as tissue oxygenation and vascular integrity [5,67,68]. That the presence or absence of AHR affects gene expression of HSCs and MPPs indicates that it plays a role in the production of hematopoietic cells. This is highly relevant to toxicology and pharmacology. For instance, AurA is already a promising cancer therapy target (reviewed in [69]). Therefore, targeting the AHR-AurA and, more broadly, the AHR-polo-like kinase axis, provides an exciting new avenue for treating hematologic diseases. Given that AHR also affects proliferation of other cell types, it is possible that AHR-AurA connections contribute to AHR's influence on proliferation of non-hematopoietic cells in peripheral tissues, such as the mammary gland, skin, and nervous systems [70–72].

Moreover, these new findings point to a broader role of AHR in other types of progenitor cells, such as those in the skin, liver, and GI tract, which have high levels of AHR, are sensitive to modulation by exogenous AHR ligands, and in which AHR signaling has been associated with altered proliferation [61,66,71,73]. Additionally, depending on context, exogenously provided AHR ligands can exacerbate or attenuate disease by modulating cell proliferation and differentiation [71,74,75]. That AHR can promote or dampen cellular processes may seem paradoxical. Yet when contextualized with the transcriptomic findings presented herein, suggests AHR helps cells sense and respond to external cues and integrate their responses within the complex regulatory networks that control cellular function. From this perspective, AHR signaling appears to assist in the ability of cells to sense and respond to environmental cues that modulate hematopoietic stem and progenitor cells, and provides insight into how AHR may affect other cells throughout the body.

**Supplementary Materials:** The following supporting information can be downloaded at: https://www.mdpi.com/article/10.3390/receptors2010006/s1, Figure S1: Flow cytometry gating used in isolation of HSCs and MPPs for RNA-Sequencing; Figure S2: Flow cytometry gating strategy to measure AurA levels in hematopoietic stem and progenitor cells; Figure S3: BrdU incorporation of WT and AHR KO MPP subsets; Table S1: Genotyping PCR primers used in study; Table S2: Information on antibodies used for flow cytometry antibodies in study; Table S3: DEGs identified from analysis of WT versus AHR KO HSCs; Table S4: DEGs identified from analysis of WT versus AHR KO MPPs; Table S5: DEGs identified from analysis of WT HSCs versus WT MPPs; Table S6: DEGs identified from analysis of AHR KO HSCs versus WT MPPs; Table S7: Canonical pathways analysis identified between WT HSCs versus WT MPPs; Table S8: Canonical pathways analysis identified between AHR KO HSCs versus WT MPPs.

**Author Contributions:** Conceptualization, A.M.F., K.P.S., T.A.G. and B.P.L.; methodology, A.M.F., K.L.V., S.B., K.P.S.; software, A.M.F., S.B.; validation, A.M.F., S.B.; formal analysis, A.M.F.; investigation, A.M.F.; resources, T.A.G. and B.P.L.; data curation, A.M.F., S.B.; writing—original draft preparation, A.M.F.; writing—review and editing, A.M.F., K.L.V., S.B., T.A.G. and B.P.L.; visualization, A.M.F., S.B.; supervision, A.M.F., T.A.G. and B.P.L.; project administration, T.A.G. and B.P.L.; funding acquisition, T.A.G. and B.P.L. All authors have read and agreed to the published version of the manuscript.

**Funding:** This work was supported by grants from the National Institutes of Health (R01ES023260, R01ES004862, P30ES01247 and T32ES007026).

**Institutional Review Board Statement:** The animal study protocol was approved by the Institutional Review Board (or Ethics Committee) of University of Rochester (protocol code 2006-078E). for studies involving animals.

**Informed Consent Statement:** Not applicable.

**Data Availability Statement:** The authors confirm that the data supporting the findings of this study are available within the article or its supplementary materials. The gene expression data that support the findings of this study are openly available in the Gene Expression Omnibus, series GSE163284. All other data that support the findings of this study are available on request from the corresponding author, A.M.F.

**Acknowledgments:** The authors thank John Ashton and the staff at the University of Rochester (UR) Genomics Research Center, and Timothy Bushnell, Matthew Cochran, and the staff of the UR Flow Cytometry Core for assistance in with cell sorting and RNA sequencing.

**Conflicts of Interest:** The authors declare no competing financial interests.

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
