# Peer review of "Impact of the Aryl Hydrocarbon Receptor on Aurora A Kinase and the G2/M Phase Pathway in Hematopoietic Stem and Progenitor Cells"

_2813-2564, doi:10.3390/receptors2010006_

Round 1
Reviewer 1 Report
In this work, the authors identified a novel mechanism through which the aryl hydrocarbon receptor (AHR) controls the proliferation and differentiation of hematopoietic stem cells. Specifically, by conducting flow cytometric analyses, the authors found that stem cells isolated from AHR-deficient mice proliferate faster than those from their wildtype littermates. RNAseq analyses aiming to identify underlying alterations on the transcriptome level and subsequent mechanistic studies revealed that AHR deficiency/antagonism is associated with an increased expression of Aurora A kinase which is known to control G2/M transition. Overall, I think the topic of the manuscript is timely and very interesting. It provides a potential novel mechanism through which AHR affect stem cell homeostasis and fate.
The major point of concern is that the connection between elevated Aurora A levels and increased proliferation rate of AHR deficient cells is largely descriptive. Experiments providing evidence that the elevated levels of Aurora A are in fact responsible for the increased division of AHR-deficient stem cells are missing. Given that interactions of AHR with several other cell-cycle regulators, including RB, E2F, p21 etc., have been described in literature, the authors should include a respective functional (in vitro?) experiment.
Minor points:
1. The contrast of the immunoblot shown in figure 4A is very high. The authors should reduce contrast to make the background of the membrane visible.
2. The data indicate that AHR represses the expression of AURKA in stem/progenitor cells. Do the authors have any idea regarding the underlying molecular mechanism? Is there anything known about the major transcriptional regulators of AURKA gene and their potential interaction with AHR-dependent signaling pathways? I would recommend to include at least a few sentences about this in the discussion section.
Author Response
A: We agree with the reviewer’s sentiment, and acknowledge that future studies to pinpoint interactions of AHR with AurA, and with other cell cycle regulators, will be of great interest and value to the fields of AHR and stem cell biology. Within our revised paper, we have expanded the scope of the Discussion section to recognize this more clearly, and provide some plausible mechanisms by which the AHR-AurA relationship can be explained (pg. 15-17). Our findings show that there is an inverse relationship between levels of between AHR and AurA, and we agree that this does not preclude AHR mediated regulation of other cell cycle regulators. We hope that sharing this new finding opens up multiple avenues for follow-on studies by us and by others, including in vitro experiments which may build on this in future studies.
Minor points:
- The contrast of the immunoblot shown in figure 4A is very high. The authors should reduce contrast to make the background of the membrane visible.
A: At the reviewer’s suggestion, we have reduced the contrast of the image presented in Figure 4A to make the background visible. We also revised the Figure legend to make it clear to readers that this is not an immunoblot, but is a gel showing the results of PCR.
- The data indicate that AHR represses the expression of AURKA in stem/progenitor cells. Do the authors have any idea regarding the underlying molecular mechanism? Is there anything known about the major transcriptional regulators of AURKA gene and their potential interaction with AHR-dependent signaling pathways? I would recommend to include at least a few sentences about this in the discussion section.
A: We thank the reviewer for this comment and suggestion. We have updated the Discussion section (pp. 16-17) to expand the scope of the discourse and have provided relevant references. Briefly, there are several possible ways in which AHR could influence AurA, It is plausible that AHR binds AHREs to act as an transcriptional regulator directly, or acts as an enhancer, an insulator, or facilitates the formation of a chromatin loop between two AHRE sites. Figuring this out will be experimentally complex as the full slate of the transcriptional regulators of the Aurka gene is an active area of research, and complex. The complexity may in part be due to the fact that regulators are cell-type-specific.
Reviewer 2 Report
The article by Franchini et al investigates mechanism by which AhR regulates HSC homeostasis and HSPC differentiation potentially via AurA. Authors utilize flow cytometry and transcriptomics approach in HSC and MPP cells isolated from control and AhR iKO mice to identify impact of AhR on Aurora A kinase in G2/M phase pathway in HSPCs.
1) Although, PCR has confirmed AhR knockdown - a genetic and pharmacological (by CH-223191 antagonist) inhibition of AhR at protein level should be shown as a control.
2) Please include name and reference to a pathway analysis program utilized.
3) Although a clear negative correlation between AhR and AurA has been established by the present study – a potential mechanism by which absence of AhR regulates AurA and G2/M phase pathway should be discussed in details in the discussion section (direct regulation of Auraka by AhR, AhR-AurA protein/pathway interaction and/or other mechanisms).
Author Response
The article by Franchini et al investigates mechanism by which AhR regulates HSC homeostasis and HSPC differentiation potentially via AurA. Authors utilize flow cytometry and transcriptomics approach in HSC and MPP cells isolated from control and AhR iKO mice to identify impact of AhR on Aurora A kinase in G2/M phase pathway in HSPCs.
- Although, PCR has confirmed AhR knockdown - a genetic and pharmacological (by CH-223191 antagonist) inhibition of AhR at protein level should be shown as a control.
A: We completely agree that it is ideal to have gene expression and protein level data. We have previously used several different commercially available antibodies for examining AHR protein using immunoblotting. However, those particular products are no longer available, and we have not been satisfied with the quality and specificity of current anti-AHR antibodies. We have used several recent products. Therefore, we did not add immunoblotting this our current paper. Using PCR, our data are consistent with the excision of AHR with other Cre-LoxP models, for example: JA Bennett et al., 2018; GB Jin et al., 2014.
- Please include name and reference to a pathway analysis program utilized.
A: We have amended the methods to indicate usage of Ingenuity Pathway Analysis as the source of our pathways analysis and have referenced it accordingly.
- Although a clear negative correlation between AhR and AurA has been established by the present study – a potential mechanism by which absence of AhR regulates AurA and G2/M phase pathway should be discussed in details in the discussion section (direct regulation of Auraka by AhR, AhR-AurA protein/pathway interaction and/or other mechanisms).
A: To address this suggestion, we have substantially revised and expanded the Discussion section (p. 15-16) to describe possible mechanisms by which AHR could regulate Auka levels, and more information about AurA and regulation of proliferation.
Reviewer 3 Report
The submitted manuscript aims to examine the role of the aryl hydrocarbon receptor on the hemapoietic stem cell (HSC) and multipotent progenitors cell (MPP) populations. Using an AHR knockout model and RNA sequencing on the HSC and MPP populations the authors identify the Aurora A Kinase pathway among the genes showing differential expression between HSC and MPP cells in the knockout model only. Using flow cytometry, they further show that Aurora A kinase protein levels are increased in these cell populations using three different models of AHR ablation/inhibition. Collectively, the authors conclude that dysregulation of the Aurora A Kinase could promote cell proliferation when AHR is repressed. The findings presented in the manuscript would be of interest to readers of Receptors and it is recommended that it be accepted pending minor revisions.
· Figures – Throughout the manuscript it is not immediately clear what the units of the data presented is for the gene expression plots. For example, Figure 1D shows a scale from 0.68 to 1 but it is not noted anywhere what this means (is it correlation coefficient?). Similarly, what are the units for the scale on Figure 1F? In other cases it is hard to locate. For example Figure 1B indicates the units while one would intuitively look at the scale for the units. Figure 1C only states the units in the text.
· More information is needed to describe the Hierarchical clusters (Figure 1D). How was it performed? It looks like it is clustering of correlation coefficients rather than clustering based on the gene expression values.
· More information is needed about the elimination of outliers mentioned in the materials and methods. What thresholds were used for excluding outliers? Given that “extreme outliers by hierarchical clustering” were removed how could hierarchical clustering not confirm delineation of HSCs and MPPs as stated in the results section?
Author Response
A: We thank the reviewer for their feedback.
Figures – Throughout the manuscript it is not immediately clear what the units of the data presented is for the gene expression plots. For example, Figure 1D shows a scale from 0.68 to 1 but it is not noted anywhere what this means (is it correlation coefficient?). Similarly, what are the units for the scale on Figure 1F? In other cases it is hard to locate. For example, Figure 1B indicates the units while one would intuitively look at the scale for the units. Figure 1C only states the units in the text.
A: This feedback is helpful. We have made certain that all figure panels have axis labels, scaling is clearly defined, and units have been added to metrics that have units. Axis labels on graphs have been improved and we hope that what each graph conveys is clearer.
More information is needed to describe the Hierarchical clusters (Figure 1D). How was it performed? It looks like it is clustering of correlation coefficients rather than clustering based on the gene expression values.
A: We have revised the second paragraph of the Results section (pp. 10-11) to make this clearer, and also updated legend for Figure 1. Briefly, the heatmap shows hierarchically clustered Euclidean distances between samples from the regularized log transformation of the normalized count data shows sample grouping by cell types. The darker shades of blue indicate sample distance and higher relatedness between samples (libraries).
More information is needed about the elimination of outliers mentioned in the materials and methods. What thresholds were used for excluding outliers? Given that “extreme outliers by hierarchical clustering” were removed how could hierarchical clustering not confirm delineation of HSCs and MPPs as stated in the results section?
A: We revised the methods section about RNAseq and transcriptome analysis (p. 8) to clarify this. Briefly, one of the samples was removed from the differential expression and downstream analyses due to it having much lower mapping quality control metrics. This was validated by the singular dataset being placed away from all other samples in Principal Component Analysis (PCA) as well as unsupervised hierarchical clustering using all genes.
Reviewer 4 Report
The manuscript by Franchini A.M. et al examines function of the AHR in differentiation of hematopoietic stem and multipotent progenitor cells. The topic fits well with the scope of the journal “Receptors”, and results presented by the authors support their conclusion. Below are a few questions raised while reading this manuscript:
1. In the DEG data that the authors presented for comparisons between WT and AHR KO (supplementary table S3 and S4), the gene Akr1c13 was also shared between these two databases. Please also indicate supplementary table S4 in the main text when appropriate.
2. The authors concluded that AHR regulates the transition from HSC to MPP as many more DEGs were identified than in the direct comparison between WT and AHR KO HSC and/or MPP. If the conclusion was correct, should DEGs identified from the direct comparison between WT and AHR KO MPP provide the same conclusion? Why there are so little difference between WT and KO MPP?
3. In the middle of this study, the authors started suddenly to examine Aurora A expression in MPP subpopulations. Please provide the logic to study subpopulations, why it’s needed at this point? Why the authors did not provide similar information in Figure 1, meaning the percent and BrdU+ MPP subpopulations?
4. The authors showed that AHR antagonist did not alter the frequency of HSC and MPP and AurA+ HSC and MPP in bone marrow but did increase the AurA expression in these cells. Is this observation consistent with what has been shown before? Please provide discussion.
5. Does AHR regulate the G1/S and G2/M checkpoints in other type of stem and/or progenitor cells? As the authors conclude that the AHR regulates differentiation of HSC, a discussion comparing different types of stem and progenitor cells will be helpful.
Author Response
- In the DEG data that the authors presented for comparisons between WT and AHR KO (supplementary table S3 and S4), the gene Akr1c13was also shared between these two databases. Please also indicate supplementary table S4 in the main text when appropriate.
A: We have updated the Results section of the main text (p. 11) to reflect that Akr1c13 appears in Supplemental Tables S3 and S4.
- The authors concluded that AHR regulates the transition from HSC to MPP as many more DEGs were identified than in the direct comparison between WT and AHR KO HSC and/or MPP. If the conclusion was correct, should DEGs identified from the direct comparison between WT and AHR KO MPP provide the same conclusion? Why there are so little difference between WT and KO MPP?
A: Thanks for these questions, as we have thought about them too. We have expanded Figure 1 to include data showing that AHR KO mice also have a slightly higher frequency of MPPs compared to WT mice, but not a significantly greater proportion of them were proliferating (i.e. were BrdU+). These observations, combined with the limited number of DEGs when we compared the same cell type (either HSCs or MPPs) from WT vs AHR KO mice, and the large number of DEGs when we compared transcriptomic profiles of HSCs and MPPs within and then across genotype, suggest to us that AHR is involved with regulating the transitional state between HSC and MPP. This is conceptually consistent with some of the work of Ko et al., who showed that AHR must be downregulated during mitosis and differentiation of embryonic stem cells (Ko et al., 2014 and 2016). In addition to modifying Figure 1, we also expanded the Discussion to better frame our observations with this prior work.
- In the middle of this study, the authors started suddenly to examine Aurora A expression in MPP subpopulations. Please provide the logic to study subpopulations, why it’s needed at this point? Why the authors did not provide similar information in Figure 1, meaning the percent and BrdU+ MPP subpopulations?
A: This is helpful feedback. We have revised Figure 1 to include the frequency of MPPs, and the percentage of MPPs that are BrdU+, and updated the text in the Results section (p. 10) and the Figure 1 legend. We have also included Supplemental Figure S3, which shows the impact of the loss of AHR MPP proliferation by populations. To summarize, there was significantly greater percentage of BrdU+ MPP1 cells (i.e., the short term HSCs) in AHR knockout mice, compared to wildtype mice.
- The authors showed that AHR antagonist did not alter the frequency of HSC and MPP and AurA+ HSC and MPP in bone marrow but did increase the AurA expression in these cells. Is this observation consistent with what has been shown before? Please provide discussion.
A: We revised the Discussion (pp. 16-17) to incorporate this feedback. Briefly summarized, this seems consistent. Although we are not otherwise aware of another study that has directly measured AurA in HSCs and MPPs for which to further compare our data to, the overall idea that AHR influences early stages of hematopoiesis is consistent. As AHR is attenuated or absent, AurA levels are elevated and there are skewed frequencies of hematopoietic stem and progenitor cell subsets.
- Does AHR regulate the G1/S and G2/M checkpoints in other type of stem and/or progenitor cells? As the authors conclude that the AHR regulates differentiation of HSC, a discussion comparing different types of stem and progenitor cells will be helpful.
A: We have revised the discussion section (pp. 15-16) to make it clearer that our findings align with prior reports of AHR and G2/M checkpoint regulation in other stem cells. Specifically, we cite work that shows the breadth by which AHR has been implicated in regulating cell cycle entry and cellular proliferation in numerous stem cells such as cardiomyocytes, embryonic, liver, and thymic stem cells.
Round 2
Reviewer 1 Report
The authors have adequately addressed my criticism, I recommend the publciation of the article in its present form.